# Life Cycle Emissions of Passenger Vehicles in China: A Sensitivity Analysis of Multiple Influencing Factors

**Haoyi Zhang** [1,2], **Fuquan Zhao** [1,2], **Han Hao** [1,2] and **Zongwei Liu** [1,2,3,*]

1   State Key Laboratory of Automotive Safety and Energy, Tsinghua University, Beijing 100084, China
2   Tsinghua Automotive Strategy Research Institute, Tsinghua University, Beijing 100084, China
3   MIT Sloan School of Management, 77 Massachusetts Ave., Cambridge, MA 02139, USA
*   Correspondence: liuzongwei@tsinghua.edu.cn

**Abstract:** To reduce greenhouse gas emissions from passenger vehicles, new energy vehicles are actively promoted by China's government. Various power system types are being developed and their sales keep increasing. However, there is uncertainty about the greenhouse gas emission of different vehicle types. This paper studies the life cycle carbon emissions of passenger vehicles in China. A calculation model is established with consideration of all types of power systems, model classes, and electric driving ranges. In order to calculate and compare the effect of carbon emission reduction on all types of vehicles, a sensitivity analysis is conducted in two ways to study three of the main influencing factors. The results show the carbon emission-reducing effect of different factors on different stages in the life cycle. It is known that different influencing factors have different effects on these stages. Since there is a variation in different vehicle types, the carbon reduction effect caused by these factors is different for these vehicle types. This paper describes a sensitivity analysis of three main influencing factors and puts forward relevant policy recommendations to reduce the carbon emissions of passenger cars during their life cycle based on these results. It is necessary to take the vehicle life cycle as a whole for carbon emission management. The conclusions of this paper can be used for vehicle manufacturers to decide the focus of technology research, and also have important reference significance for enterprises when making life cycle carbon reduction strategies for their products. It is also of certain value for China to formulate a medium- and long-term carbon emission reduction strategy for the passenger car industry.

**Keywords:** greenhouse gas emissions; new energy vehicle; life cycle assessment; power generation industry

## 1. Introduction

The development of the automobile industry has provided strong support for the sustainable development of China's economy [1]. The scale of China's automotive industry has grown rapidly in the last two decades until the COVID-19 epidemic [2]. Since the sales exceeded those in the United States in 2009 [3], China has been the largest market for automobiles for more than ten years [4]. In 2022, 26.864 million automobiles were sold according to data from the China Association of Automobile Manufacturers. Due to the huge sales volume and ownership, the environmental impacts caused by rapid vehicle growth have long been studied by academic circles [5]. Moreover, the oil external dependency of China reached 70.9% in 2022 [6], which is a major issue for the economy and national security in China. "Electrification" of the passenger vehicle is strongly promoted by the Chinese government to reduce the current high fuel consumption and high pollution. New energy vehicles are promoted by the government and the sales distribution has risen rapidly in recent years, reaching 14.8% in 2021 [7].

The life cycle emissions of various new energy vehicles are different and have not been compared quantitatively yet [8]. These unquantified differences in life cycle emissions and their reduction potential lead to confusion when the government formulates a

NEV-promoting policy and subsidises NEVs. It is also confusing for original entrusted manufacture (OEM) to decide which is a worthwhile power system in which to invest resources in research and development (R&D). Therefore, it is necessary to analyze the life cycle emission and to know which type of power system has lower carbon emissions or higher carbon reduction potential in the future.

### 1.1. Literature Review

The method of life cycle analysis has been used since the 1990s. The main research methods are rarely changed. The life cycle of the object is divided into three stages: production, use, and end-of-life stage. An inventory is established in the process and the carbon emissions corresponding to the resource consumption are analyzed separately.

At present, there are two databases used for vehicle emission calculation, established by authoritative research institutions: "Greenhouse emissions, Regulated Emissions, and Energy use in Transportation" (GREET) led by Argonne National Laboratory, which mainly collects data from the United States, and some resources and process-related data from other major regions [9]. The other database, "Ecoinvent", was established by the Swiss Centre [10]. In addition, there are many other types of life cycle research software that cover car-related research, such as "GaBi" developed by Thinkstep in Germany [11] and "SimaPro", which was developed by PRé in the Netherlands [12]. According to the life cycle assessment theory, China has also established the "China Mobile Life Cycle Database (CALCD)" based on Chinese data.

Most published life cycle research about passenger vehicles directly or indirectly cites data from the main data sources.

Weeberb J. Requia et al. calculated the daily emission reduction rate per 1% penetration of PHEVs in Canada by using the PEV-CIM model suitable for Canada's national conditions and selecting two PHEVs on sale as reference [13]. Ziyang Wu et al. selected two reference models to study the carbon emissions of single BEVs and PHEVs under the provincial power grid in China by combining the GREET model and the actual data input of Chinese automobile enterprises [14]. Linda Ellingsen and others discussed the impact of increasing battery capacity and driving range on the environment of electric vehicles, and the impact of the proportion of energy generated by the grid on the emissions of new energy vehicles throughout their life cycle by sorting out the relation between curb weight, power consumption, and battery capacity, which were not limited to the sale of a certain model and made the model have flexibility and expansibility [15].

For passenger vehicles, there have been many studies on the carbon emissions of traditional fuel vehicles and various new energy vehicles. However, these studies mainly focus on one reference vehicle model. At most, they only to compare the life cycle carbon emissions of different power types or driving ranges under the same conditions. These studies did not and were not able to compare the life cycle carbon emissions of various types of vehicles, and only discussed the changes in carbon emissions of different types of vehicles under the future technology development trend, providing a basis for selecting vehicle types and technology combinations that are conducive to carbon emission reduction. Therefore, research should be conducted to compare all types of vehicles, which also should be able to take the impact of the macro changes in various technologies on vehicle performances and carbon emissions into consideration.

In this paper, three main influencing factors are selected and studied to figure out the best technology roadmap of passenger vehicles, when only carbon emissions are considered. These selected influencing factors are shown below.

### 1.2. Important Factors Affecting Fleet Carbon Emissions

1.2.1. Emissions of the Power Grid

Coal thermal power accounts for a major part in China's electricity generation, which produces a lot of GHG emissions. China is now promoting clean energy generation. The coal–thermal–electricity ratio was reduced from 76.3% in 2010 [16] to 60.7% in 2020 [16].

Meanwhile, solar power and wind power grow rapidly, driven by supporting policy subsidies, and continuous cost. The average generation emission has been reduced accordingly. Table 1 shows the distribution of the resources used for generation in China. It can be seen that power generation of all kinds of clean energy is rising quickly if compared with thermal power. The penetration of renewable energy will continue to rise due to the continuously declining installation cost of renewable energy power generation and the goal of carbon neutrality [17].

**Table 1.** Power generation of different energy resources (100 million kWh) [18].

| Year | Thermal Power | Hydroelectric Power | Nuclear Power | Wind Power | Solar Power |
|------|---------------|---------------------|---------------|------------|-------------|
| 2011 | 38,337.0 | 6989.4 | 863.5 | 703.3 | 6.0 |
| 2012 | 38,928.1 | 8721.1 | 973.9 | 959.8 | 36.0 |
| 2013 | 42,470.1 | 9202.9 | 1116.1 | 1412.0 | 84.0 |
| 2014 | 44,001.1 | 10,728.8 | 1325.4 | 1599.8 | 235.0 |
| 2015 | 42,841.9 | 11,302.7 | 1707.9 | 1857.7 | 395.0 |
| 2016 | 44,370.7 | 11,840.5 | 2132.9 | 2370.7 | 665.0 |
| 2017 | 47,546.0 | 11,978.7 | 2480.7 | 2972.3 | 1178.0 |
| 2018 | 50,963.2 | 12,317.9 | 2943.6 | 3659.7 | 1769.0 |
| 2019 | 52,201.5 | 13,044.4 | 3483.5 | 4057.0 | 2240.0 |
| 2020 | 53,302.5 | 13,552.1 | 3662.5 | 4665.0 | 2611.0 |

### 1.2.2. Battery Technology

The power battery is one of the most important parts of the BEV. The GHG emissions of the production process are different between different types of batteries. Moreover, the emissions indirectly relate to the energy density of the batteries. With a certain electricity driving range, different energy densities will affect the loaded battery capacity and power consumption of the vehicle [19]. Since NEVs are newly implemented in China and the technology is in the early stage, the energy density of the power battery is growing fast. It mainly influences the weight of the battery carried in a vehicle and influences the emissions of battery manufacturing. Moreover, it also influences other parameters of the models, such as the electricity consumption, output power, safety, etc. In particular, the electricity consumption impacts carbon emission indirectly.

The technology of the power battery is developing fast, and advanced technology will be implemented. It is necessary to predict the roadmap of battery technology development. In this research, the batteries are roughly divided into several categories based on the materials of the electrode and the state of the electrolyte. It is known that lithium-ion battery cells achieve the highest gravimetric energy and power densities of all commercially available rechargeable batteries [20]. In addition, lithium is a metal element with the lowest molecular weight among all elements, which makes it the most advantageous element that can make the battery energy density higher. In this paper, there are two main types of batteries applied to NEVs: LFP and ternary lithium battery. Both of them are lithium-ion batteries. The ternary lithium battery has developed fast in recent years, and the mainstream cathode material is LiNixCoyMn1-x-yO$_2$, which mainly consists of metallic elements of nickel (Ni), cobalt (Co), and manganese (Mn), simplified as "NCM" [21,22]. These two types of batteries began to develop rapidly at the same time as NEVs in China, and the power density of these two kinds of batteries has risen rapidly since 2017 [23]. The energy density and its improvement should be regarded as key influencing factors and are considered in this paper to analyze the effect of batteries on life cycle emissions.

### 1.2.3. Fuel and Power Consumption

Fuel and power consumption are key factors that directly relate to the life cycle emissions of vehicles. After a long period of development, the fuel consumption of internal combustion engine vehicles (ICEVs) has hardly changed in recent years. The average fuel consumption of passenger cars in China has experienced a relatively rapid decline in recent

years, with a 2% decline every year [24]. Most of the reduction is due to the introduction of hybrid vehicles and all types of NEVs. According to the "Chinese Corporate Average Fuel Consumption and New Energy Vehicle Dual-Credit Regulation", CAFC regulations have a great preference for new energy vehicles. The calculated corporate average fuel consumption value may be far lower than the average carbon emissions of enterprises' ICEV and HEV models [25]. The actual rate of fuel consumption improvement is not so fast. Since the two types of vehicles are analyzed separately in this paper, the fuel consumption of ICEVs is assumed to have no significant change, while the fuel consumption of hybrid electric vehicles (HEVs) will have a significant change, which is an important factor affecting vehicle carbon emissions.

*1.3. Development of Driving Cycle*

There are many different operation modes to be assessed, e.g., idle, engine start, tip-in, tip-out, launch, and gear shifts [26], and various modes are used in the actual driving process. To evaluate the performance of the vehicle, it is necessary to evaluate the comprehensive performance of all modes. So, the driving cycle is used to evaluate the power consumption and fuel consumption of vehicles. The New European Driving Cycle (NEDC) has been applied in China since 2008 [27]. However, to better fit the real driving scenarios in China, two new test cycles are used, the China Automotive Test Cycle (CATC) for BEVs and Worldwide Harmonized Light Vehicles Test Cycle (WLTC) for all fuel-consuming vehicles [27]. Both new test cycles will be implemented before 2025. Since different test cycles are used for different power systems, test cycles should be unified when comparing carbon emissions.

This paper describes a sensitivity analysis with an LCA carbon emission calculation model for various kinds of passenger cars. A calculation method is set up to analyze the carbon emission of all kinds of new energy vehicles from the aspect of life cycle assessment (LCA).

The remainder of the paper is organized as follows. In Section 2, the calculation model of vehicle LCA carbon emission is introduced. In Section 3, the method and results of sensitivity analysis are described. In Section 4, the conclusion of the sensitivity analysis is presented, and a few suggestions for reducing LCA carbon emission are also given.

## 2. The Calculation Model of the LCA Carbon Emission

This paper analyzes and compares the GHG emission reduction effects in various types of vehicles caused by different key influencing factors. Firstly, the emissions should be calculated considering all these factors and all types of vehicles. A life cycle emission calculation model (LCARM) established by Zhang is used [28]. Then, sensitivity analysis of each parameter is carried out. The research is conducted as follows.

*2.1. Model Classification and Market Share Evaluation*

Influence should be measured for each type of vehicle separately. A classification of all passenger vehicles is conducted and several representative models are set. These models should be able to represent all possible vehicle types. Then, the effect on any model can be known by calculating the effect of the corresponding representative model. The following classification criteria are considered: model type, vehicle class, power system, and electric driving range. For vehicle type, sedans and sports utility vehicles (SUVs) are set. These two models are the ones with significant differences and large sales volumes in the Chinese market. Hatchback, small, or luxury cars are also considered sedans in this research. The classification of the SUV is not clear in China, but generally refers to models with higher pass ability but are designed based on the chassis of sedans, and off-road vehicles such as Jeeps are also regarded as SUVs. China has set a six-level classification based on the classification method of the German Volkswagen, which is A00, A0, A, B, C, and D. Sedans and SUVs are classified with similar criteria.

All these kinds of new power systems have been implemented in China in recent years. ICEVs are the most common power systems. Their fuel consumption improvement has been slow in recent years. After the introduction of hybrid electric vehicles (HEVs), the thermal efficiency of the engine has been rapidly improved. This power system is equipped with an internal combustion engine and electric motor. When the output power of the engine is higher than the requirements, the electric motor operates as a generator and outputs the stored power when additional power is required. This feature enables the powertrain to keep the engine running in the highest efficiency mode. In this paper, HEVs are divided into two types, namely, mild hybrid electric vehicle (MHEV) and full hybrid electric vehicle (FHEV) with a 48V electrical system. The plug-in hybrid electric vehicle (PHEV) is also composed of an engine and motor, but the voltage and power are significantly higher than that of the 48V electrical system, and the battery capacity is also higher [29]. Most importantly, PHEV batteries can be recharged from the outside. In most cases, the vehicle can run without fuel consumption, and this power system can only consume fuel in conditions of high power requirement or when the power battery capacity is low [30]. Similarly, an engine and one or more motors are equipped in extended range electric vehicles (EREVs). However, the engine is only used as a charger while the motor is the only power source in the powertrain [31].

With the development of technology, more automobile manufacturers have adopted an engine alternator with a more compact structure and smaller displacement [32,33]. Battery electric vehicles (BEVs) are only driven by electric motors. By loading batteries with different capacities, BEVs with various electric driving ranges are implemented, mostly longer than those of PHEVs or EREVs, but they are unable to move when the battery runs out. Therefore, consumers have some anxiety about mileage and pursue a longer driving range. PHEVs, EREVs, and BEVs are collectively referred to as NEVs. The carbon emissions at each stage of the life cycle are different.

The sales of NEVs has been growing fast in recent years, and the demand for battery electric vehicles (BEVs) and hybrid electric vehicles (HEVs) is believed to have increased significantly [34], caused by stricter fuel consumption legislation, government incentives, higher petrol price, and policies at the use stage [35]. New energy vehicles are divided into BEVs and PHEVs according to the powertrain when making statistics on sales volume. An EREV is regarded as a PHEV while the HEV is not regarded as a new energy vehicle in the fuel consumption restriction regulations in China. At present, the BEV plays a major role in the sales of new energy vehicles, 81.8% in 2021 according to the China Passenger Car Association [36]. The EREV is newly implemented in China and only a few models are on sale.

### 2.2. LCA Carbon Emission Calculation

The sensitivity analysis is conducted based on an established calculation model, "LCARM", and the life cycle calculation method established by Zhang et al. [28]. In this model, GHG emissions of vehicles in all stages of the life cycle are calculated with consideration of various influencing factors. Parameters related to vehicle technical indicators include curb weight, fuel consumption, power consumption, electric driving range, and battery energy density. Other parameters include the emission coefficient of the power grid, emissions during vehicle and battery manufacturing, and end-of-life process. These factors interact with each other, resulting in a complex mechanism between these factors and the life cycle emissions. Among them, the interaction mechanism of the battery capacity, battery power density, battery weight, curb weight, power consumption, and electric driving range is particularly necessary.

The calculation method of the life cycle emissions for each vehicle can be described as Equation (1).

$$E_{LCi} = (E_{VMi} + M_{Bi} \times E_{BM}) + [\{EC_i \times GE \times UF + FC_i \times FE \times (1 - UF)\} \times R_u] + (E_{VEi} + M_{Bi} \times E_{BE}) \qquad (1)$$

where $E_{LCi}$ is life cycle emissions of vehicle model $i$; $E_{VMi}$ is emissions of vehicle manufacturing of vehicle model $i$; $E_{BM}$ is emissions during battery manufacturing ($gCO_{2e}/kg$); $EC_i$ is the electricity consumption of model $i$ (kWh/100 km); $GE$ is power grid carbon emission (kg/kWh); $FC_i$ is the fuel consumption of model $i$ (L/100 km); $FE$ is the life cycle emission of fuel consumption ($CO_{2e}/L$); $R_u$ is the driving mileage of the vehicle during the use stage (100 km); $E_{VEi}$ is emission in the end-of-life stage of vehicle model $i$; $M_{Bi}$ is the battery weight of model $i$; $E_{BE}$ is emissions caused by the battery installed in vehicle model $i$ during the end-of-life stage ($gCO_{2e}/kg$).

In this study, to calculate the power consumption and fuel consumption of PHEVs and EREVs, the ratio between CS and CD conditions when driving should be clear. A utilization factor (UF) is introduced.

In this paper, vehicles with the same power systems but different driving ranges are considered to have the same chassis but carry different amounts of batteries. In fact, the battery is not proportional to the driving range. Therefore, the battery carrying capacity of different vehicle types is calculated by the following method.

The battery capacities for different representative models in this research are determined by the given driving ranges, and the weight can be calculated by average battery energy density, as shown in Equation (2). The vehicles are assumed to consist of two parts, the part with the battery and the part without the battery, and the curb weight of a vehicle can be calculated using Equation (3):

$$M_{Bi} = C_i \times \omega \tag{2}$$

$$M_i = M_{Bi} + M_{Ri} \tag{3}$$

$M_i$ is the curb weight of vehicle $i$; $M_{Ri}$ is the weight of the part excluding the battery of vehicle $i$; $C_i$ is the battery capacity of model $i$; $\omega$ is the predicted average battery density.

The relationship between power consumption and curb weight is the key influencing factor in carbon emission calculation, especially in the usage stage. Since a characteristic of the PHEV, EREV, and BEV is that the wheel is directly driven by an electric motor, the power consumption and curb weight can be regarded as linear statistically. Data from all vehicle models sold in 2020 are collected and the fitting results are shown as Equations (4) and (5):

$$EC_i = a_j M_i + b_j \tag{4}$$

$$FC_i = c_j M_i + d_j \tag{5}$$

$a_j$, $b_j$, $c_j$, and $d_j$ are parameters in the linear fitting results. A linear fitting is conducted where all NEV passenger car models are included in all 12 from the "Catalogue of New Energy Vehicle Models Exempted from Vehicle Purchase Tax" released by the Ministry of Industry and Information Technology of the People's Republic of China in 2020 [37]. These four parameters for PHEVs are 0.011, −0.6881, 0.0044, and −2.6256, and for EREVs they are 0.0149, −8.7716, 0.0053, and −4.6469. $a_j$ and $b_j$ for BEVs are 0.0044 and 5.7847.

With these known factors and equations, the battery capacity of all representative NEV models can be obtained, as shown in Equation (6). Then, all other parameters needed can be referred to with these equations after the capacities are found.

$$C_i = \frac{(a_j M_r + b_j) ER_i}{(1 - \omega \cdot ER_i)} \tag{6}$$

where $C_i$ is the battery capacity of model $i$; $ER_i$ is the electric driving range of model $i$.

Then, the carbon emission of different models in the usage stage is calculated based on parameters of power and fuel consumption. The carbon emission of manufacturing and end-of-life stages is calculated based on the weight and capacity of the battery, and the weight of the rest. With other relevant parameters known, the emissions in each stage of

the life cycle of all representative models can be obtained. Finally, the sensitivity analysis of each key factor can be conducted.

### 2.3. Sensitivity Analysis of Different Influencing Factors

There are many factors that influence the carbon emissions of vehicles, and the effects of these factors vary. So, it is important to quantify the influence of each factor and compare them. The results will help predict the trend of carbon emission development in the future with various influence factors in technological improvement.

In this research, sensitivity analysis is conducted with the three most important selected factors, which are fuel and power consumption improvement, the energy density of the battery, and the power grid emission. These three factors are all key factors that affect the LCA of emissions of nearly all types of power systems, and these are also factors that have changed dramatically in recent years. The importance of fuel economy and power consumption is highly regarded, resulting in an advance in energy-saving technology and a decline in average fuel or power consumption. Since NEVs are newly implemented in China and the technology is in the early stage, the power consumption technology has improved quickly compared to fuel consumption. Similarly, the energy density of the power battery is growing fast, and higher energy density has multiple influences on many other parameters of vehicles, with direct and indirect influence the LCA of emissions. The power grid emission is also improving fast in China, for the distribution of different generations is changing rapidly.

In this research, a sensitivity analysis is conducted to develop the potential of carbon emission reduction with these factors. It is important to analyze the relationship between the change in each influencing factor and carbon emission reduction. Additionally, a comparison is made to analyze the emissions reduction caused by each factor with consideration of their improvement in this period. Therefore, the sensitivity analysis is carried out in two ways. A separate analysis of different factors is based on the current status and a comparative sensitivity analysis compares these three factors under the same criterion. For the separate sensitivity analysis, this research focuses on the effect of each factor in 2030, because the trend of these factors is difficult to predict accurately. For the comparative sensitivity analysis, the results of carbon emissions from 2020–2030 are used, to identify the most critical influencing parameter.

### 2.4. Data Input and Acquirement

This research uses the data from Zhang's research, and base models are selected from the car market in China. This paper lists the key data and assumptions used. Representative models are set and the parameters of these models are shown in Table 2. Data of vehicle model parameters are publicly available in the government database "Inquiry of automobile energy consumption in China", established by the Ministry of Industry and Information Technology [37].

The New European Driving Cycle (NEDC) was implemented in 1997, and it is widely used in Europe and China for the fuel consumption and power consumption test cycles of passenger cars. However, it is known that the NEDC does not represent the real driving behavior of vehicles in actual traffic and, thus, does not accurately reflect pollutant emissions and fuel consumption [38]. In addition, Europe and China have been in the process of adjusting the implementation conditions in recent years, so comparing fuel or power consumption data under different conditions is not rigorous. A unified test cycle is used in this research to compare fuel consumption and power consumption of various power systems. The China Automotive Test Cycle (CATC) is the most suitable cycle, for it has been developed and implemented in China since May 2020 and will become a mandatory standard for vehicles with all types of power systems after 2025. To simplify, the power consumption under the NEDC condition can be multiplied by a conversion factor of 98.31% to obtain the power consumption under the CATC condition [39]. Similarly, the conversion factor of fuel consumption is 114% [40].

**Table 2.** Parameters of base models. (**a**) Parameters of ICEVs, (**b**) parameters of BEVs, (**c**) parameters of PHEVs, (**d**) parameters of EREVs.

(**a**)

| Type | Class | Curb Weight (kg) | Fuel Consumption (L/100 km) |
|---|---|---|---|
| Sedan | A00 | 930 | 5.4 |
| | A0 | 1150 | 5.5 |
| | A | 1340 | 5.8 |
| | B | 1540 | 6.3 |
| | C | 1855 | 7 |
| SUV | A0 | 1310 | 6.1 |
| | A | 1550 | 6.6 |
| | B | 1855 | 7.5 |
| | C | 2005 | 7.7 |

(**b**)

| Type | Class | Electric Driving Range (km) | Battery Capacity (kWh) | Battery Weight (kg) | Curb Weight (kg) | Weight Excluding Batteries (kg) | Power Consumption (kWh/100 km) |
|---|---|---|---|---|---|---|---|
| Sedan | A00 | 301 | 32.2 | 230 | 1180 | 950 | 10.7 |
| | A0 | 302 | 35.2 | 251 | 1340 | 1089 | 11.7 |
| | A | 460 | 58.8 | 354 | 1625 | 1271 | 12.8 |
| | B | 450 | 71 | 450 | 1865 | 1415 | 15.8 |
| | C | 490 | 74 | 530 | 2130 | 1600 | 15.1 |
| SUV | A0 | 415 | 48 | 353 | 1430 | 1077 | 11.6 |
| | A | 353 | 52 | 374 | 1635 | 1261 | 14.7 |
| | B | 520 | 73 | 425 | 1900 | 1475 | 14 |
| | C | 425 | 74 | 530 | 2387 | 1857 | 17.4 |

(**c**)

| Type | Class | Electric Driving Range (km) | Battery Capacity (kWh) | Battery Weight (kg) | Curb Weight (kg) | Weight Excluding Batteries (kg) | Power Consumption (kWh/100 km) | Fuel Consumption (L/100 km) |
|---|---|---|---|---|---|---|---|---|
| Sedan | A | 66 | 11 | 116 | 1650 | 1534 | 16.7 | 4.6 |
| | B | 60 | 11 | 116 | 1810 | 1694 | 18.3 | 5 |
| | C | 95 | 18 | 124 | 2005 | 1881 | 18.9 | 6.4 |
| SUV | A0 | 62 | 11 | 116 | 1548 | 1432 | 17.7 | 4.5 |
| | A | 70 | 17 | 139 | 1881 | 1742 | 24.3 | 5.2 |
| | B | 81 | 17 | 165 | 2250 | 2085 | 21 | 7.5 |
| | C | 50 | 12 | 114 | 2351 | 2237 | 24 | 6.7 |

(**d**)

| Type | Class | Electric Driving Range (km) | Battery Capacity (kWh) | Battery Weight (kg) | Curb Weight (kg) | Weight Excluding Batteries (kg) | Power Consumption (kWh/100 km) | Fuel Consumption (L/100 km) |
|---|---|---|---|---|---|---|---|---|
| SUV | A | 150 | 30 | 232 | 1940 | 1708 | 20 | 4.9 |
| | B | 150 | 35 | 247 | 2230 | 1983 | 23.3 | 6.9 |
| | C | 148 | 38.5 | 261 | 2300 | 2039 | 26 | 8.8 |

Parameters of power consumption, fuel consumption, and power battery performance are cited from "Technology Roadmap for Energy Saving and New Energy Vehicles 2.0" (hereinafter referred to as "the road map 2.0") [23], as shown in Table 3. In this table, high power consumption and fuel consumption improvement rates mean the consumption is reduced, and high battery energy density improvement rates mean the energy density is raised compared with five years ago.

**Table 3.** Improvements in key performance (the battery energy density stands for battery system).

| | BEV | | | | PHEV and EREV | | | | |
| --- | --- | --- | --- | --- | --- | --- | --- | --- | --- |
| | Power Consumption (kWh/100 km) | | | Battery Energy Density (Wh/kg) | Power Consumption (kWh/100 km) | | | Fuel Consumption (L/100 km) | Battery Energy Density (Wh/kg) |
| Class | A00 | A | B | All | A00 | A | B | All | All |
| 2025 | 90.0% | 91.7% | 92.9% | 146.3% | 90.0% | 91.7% | 92.9% | 95.6% | 130.6% |
| 2030 | 94.4% | 95.5% | 96.2% | 119.2% | 94.4% | 95.5% | 96.2% | 93.0% | 125.1% |

Other related parameters are as follows: life cycle GHG emission of fuel consumption is 91.2 g $CO_2$/MJ [41,42], the density of gasoline is 0.732 kg/L, and the gasoline calorific value is 43,070 kJ/kg. Ru is the mileage of the vehicle during the usage stage. It is considered to be 117,780 km during the 10-year service life of the vehicle in this study [43].

The UF is stipulated by the US SAEJ1711-2010 regulations [44]. It is the proportion of the driving distance in charge-depleting mode to the total distance from a fleet perspective [45]. Assuming that PHEVs and EREVs are charged once a day, the UF parameter is related to the driving range [46]. According to the driving range and UF curve, the UF is 79.30% for PHEV50, 90% for PHEV80, 95% for EREV150, and 98% for EREV200 [47].

The basic factors of carbon emissions in the manufacturing and end-of-life stage are as shown in Table 4 (taking the A-class sedans and SUVs as an example).

**Table 4.** Basic factors of carbon emissions in the manufacturing and end-of-life stage.

| Stage | Class | ICEV (t/veh) | MHEV (t/veh) | FHEV (t/veh) | PHEV Excluding Batteries (t/veh) | BEV Excluding Batteries (t/veh) | EREV Excluding Batteries (t/veh) | NMC Batteries | LFP Batteries |
| --- | --- | --- | --- | --- | --- | --- | --- | --- | --- |
| Manufacturing stage | Sedan A | 6.50 | 6.60 | 6.80 | 6.80 | 8.90 | - | 0.11 (tCO_2/kWh) | 0.11 (tCO_2/kWh) |
| | SUV A | 7.52 | 7.63 | 7.87 | 7.87 | 8.83 | 5.99 | | |
| End-of-life stage | Sedan A | 0.50 | 0.50 | 0.50 | 0.50 | 0.51 | - | 10.97 (tCO_2/t) | |
| | SUV A | 0.58 | 0.58 | 0.58 | 0.57 | 0.51 | 0.56 | | |

Many technological advancements can affect the life cycle emissions of passenger vehicles. Among them, carbon emission from power generation is considered one of the most critical parameters. China is now promoting clean energy power generation. The coal–thermal–electricity ratio was reduced from 76.3% in 2010 [16] to 60.7% in 2020 [16]. Data in Zhang's research model [28,48,49] are shown in Table 5.

**Table 5.** Power grid data and prediction.

| | Year 2020 | Year 2025 | Year 2030 |
| --- | --- | --- | --- |
| Grid carbon emission (gCO_2/kWh) | 550.7 | 463.1 | 381.8 |

To take the impact of grid carbon emissions on the production stage into consideration, the proportion of carbon emissions caused by electricity consumption in the manufacturing stage is used, as shown in Table 6.

**Table 6.** The proportion of carbon emissions caused by the power grid during vehicle (excluding battery) or battery manufacturing [50,51]. (Based on the carbon emission data in 2020).

| | Emissions of Vehicles (ICEV/HEV/PHEV/EREV) | Emissions of Vehicles (BEV) | Battery (LFP) | Battery (NCM) |
| --- | --- | --- | --- | --- |
| Proportion | 24.4% | 26.7% | 85.9% | 83.0% |

The proportion data for the year 2020 are from the "Statistics and Data Center of China Electricity Council" [52]. The predicted proportion data for the year 2030 are from the

"China Energy Outlook 2030" [53]. Carbon emission data of thermal power generation are from the "Annual Development Report of China's Electricity Industry" [52]. Carbon emission data of renewable energy power generation carbon emissions are from studies by Jiang et al. and Yu et al. [48,49]. All data for the year 2025 are the average of the 2020 and 2030 data. Grid average carbon emission is calculated considering transmission loss of 5.60% [52].

With data from the base models, power grid carbon emission, energy consumption, battery performance, carbon emission factor in three stages, and other relevant data, the carbon emission of all representative models can be calculated. The content above is the main component of the LCARM method. Except for the results of the LCA of carbon emission, all input data of the LCARM can be flexibly modified, enabling prediction and sensitivity analysis considering any single factor or combination of factors.

## 3. Sensitivity Analysis of Different Influencing Factors

Before studying the three main influencing parameters, we note that there are other parameters that may affect the calculation results of carbon emissions. These parameters neither have a certain development trend with the progress of technology, nor are widely distributed among different models in the market. However, when the average value changes, the result will still be different from the real situation. These parameters are the weight of vehicles and total driving mileage in the usage stage. The sensitivity of these two parameters is analyzed. For vehicle weight (excluding battery), the current value is a model with relatively moderate technical level in each powertrain type and model level as a reference parameter, but, in fact, the actual average value of the vehicle weight (excluding battery) in China may be different from the reference model, and the vehicle lightweighting technology also has a certain impact on the vehicle weight. This paper takes an A-class SUV as an example, and it is found that when the curb weight is 5% lower than the data used in this research, the carbon emissions of PHEV50, EREV200, and BEV300 will be 1.92% 4.47%, and 1.36% lower. Since the average weight of each vehicle class is unlikely deviate too much (considered to be within 5%), the error caused by this factor is tolerable. Total driving mileage of a vehicle directly influences the emissions in the usage stage. Although the mileage of each vehicle will be different, the average mileage is not considered to have a large deviation. If the average value of the total mileage is 5% lower, the emissions of PHEV50, EREV200, and BEV300 are 3.10%, 2.70%, and 1.84% lower. This error is tolerable when the deviation of average mileage is 5% at most.

### 3.1. Sensitivity Analysis of the Average Power Grid Emission

The influence of the power grid emissions during the manufacturing and usage stage is considered. The power grid emission is reducing because the share of clean energy in the grid is rising rapidly. The power grid emission directly influences the emission of power consumption but does not influence the basic parameters of representative models, so the relationship between power grid emission and LCA-based emission is linear.

The power grid emission is reducing quickly, with the reduction predicted to be over 30% from 2020 to 2030. In addition, it should be known that China proposed a carbon neutrality target in 2020, but to date there has been no plan for China's power grid development to target carbon neutrality. Since the data on the power grid used in this research are based on research or government plans before the carbon neutrality target was put forward, the actual situation will be much better before 2030. Therefore, it is presumed that the emission of the power grid will be 10% lower. The difference in carbon emission caused by this factor is shown in Figure 1a. Additionally, to further analyze the effect of power grid emission in each stage, a stacked column chart, shown in Figure 1b, is used to show the carbon reduction effect in different stages. The stacked bars show the amount of carbon emission reduction ($\Delta CO_2$) in different stages after the power grid emission is improved by 10%, and the percentages marked on the bars refer to the carbon emission reduction rate of the stage after the grid emission is improved by 10%, compared

to the carbon emission of the same stage in 2030. These results can be used to quantify the effect of the grid emission being further reduced. The representative A-class SUVs are selected because all types of power system can be implemented in this kind of chassis and a comparison can be conducted. Each set of bar charts compares the carbon emissions of a vehicle model under the base grid emissions and 10% lower grid emissions.

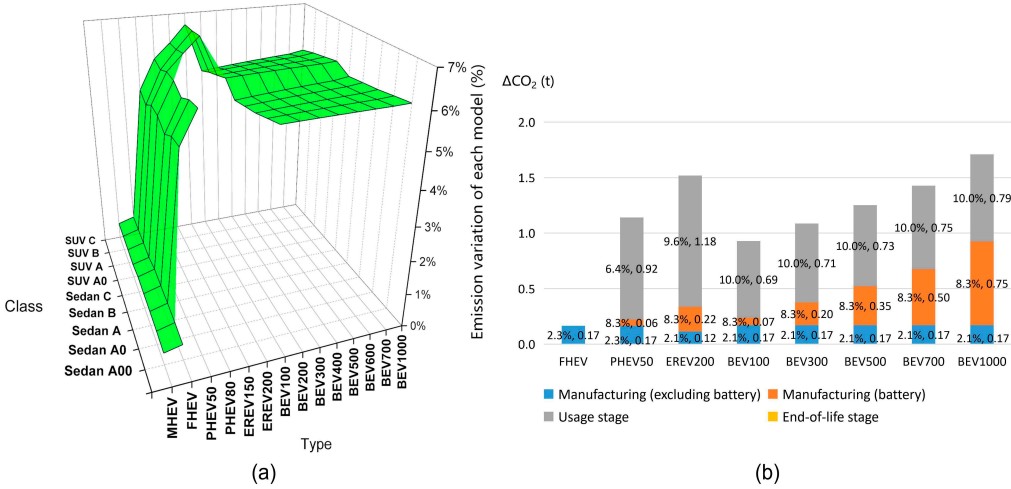

**Figure 1.** The influence of power grid emission on each type of vehicle. (**a**) Sensitivity analysis of power grid emission. (**b**) The emission distribution of different sources (A-class SUV). The numbers are emission reduction amount according to $\Delta CO_2$ (t) and the percentage is the rate of emission in the corresponding stage (reduced carbon emissions/original carbon emissions).

It is obvious that there is a significant difference between FHEVs and all kinds of NEVs. The carbon emission of ICEVs is barely influenced by the power grid emission, as a reduction of only 0.17 tons of emissions occurs in the stage of manufacturing. In contrast, the ratio of carbon emission reduction of all NEVs is similar, at around 6–7%, mainly composed of carbon emission reduction in the use phase and battery production phase. For EREV200, a reduction of 1.18 t (9.6%) of emissions occurs in the usage stage and 0.22 t (8.3%) in the battery manufacturing stage. For BEV500, a reduction of 0.73 t (10.0%) of emissions occurs in the usage stage and 0.35 t (8.3%) in the battery manufacturing stage, and a 0.17 t emission reduction occurs in the manufacturing (excluding battery) stage. In general, the power grid emission has a higher effect on PHEVs and EREVs because they experience carbon reduction in both the usage stage and battery manufacturing stage, which are two main sources of power grid emissions in vehicles.

### 3.2. Sensitivity Analysis of Power Consumption

Both fuel and power consumption are improving at a quick pace. However, the fuel-saving technology will reach its limit without considering electrification. So, power consumption is regarded as a more important factor and the sensitivity analysis is conducted focusing on it. The power consumption of each NEV not only directly influences the carbon emissions during the usage stage, but also indirectly influences the emissions of all stages as higher power consumption results in a lower battery capacity requirement. This results in a nonlinear relationship between consumption and LCA-based emissions. So, two scenarios are set, which are power consumption reduced or raised by 5% based on the predicted data for 2030. Then, the difference in LCA-based carbon emissions of each representative model under different power and fuel reductions can be obtained and is shown in Figure 2.

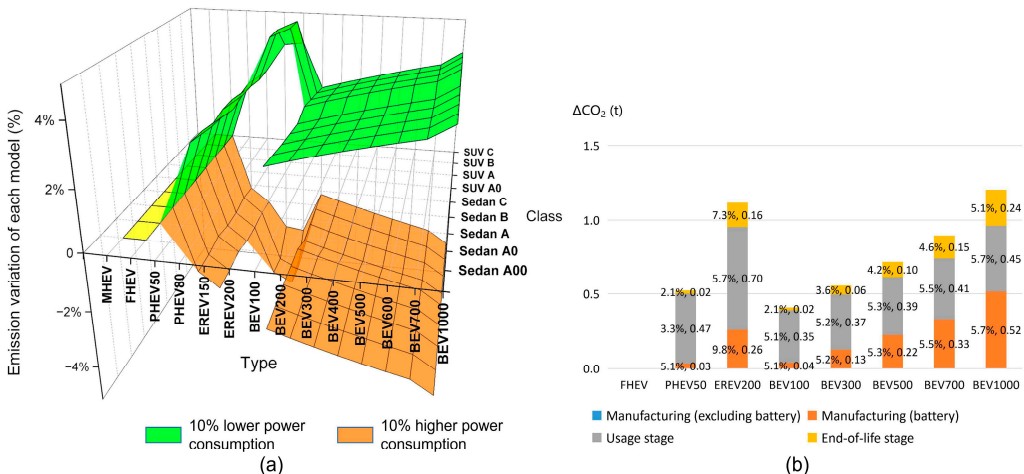

**Figure 2.** The influence of power consumption on each type of vehicle. (**a**) Sensitivity analysis of power consumption. (**b**) The emission distribution of different sources (A-class SUV). The numbers are emission reduction amount according to $\Delta CO_2$ (t) and the percentage is the rate of emission in the corresponding stage (reduced carbon emissions/original carbon emissions).

Similar to Figure 1b, the stacked bars in Figure 2b show the amount of carbon emission reduction ($\Delta CO_2$) in different stages after the power consumption is improved by 10%, and the percentages marked on the bars refer to the carbon emission reduction rate of the stage after the power consumption is improved by 10%, compared to the carbon emission of the same stage in 2030. These results can be used to quantify the effect of the grid emission when further reduced. The representative A-class SUVs are selected.

The results show that power emission benefits all power systems but FHEVs. The ratios of emission reduction of all NEVs are significantly different. In general, power consumption only influences the emission of NEVs, the increase or decrease in carbon rate is from 2–5%. When power consumption is reduced by 10%, the reduction ratio of different power systems increases with the driving range. The reduction ratio of EREV200 is the highest, even higher than that of BEV200. The main reason for this phenomenon is that the curb weight of this power system is relatively higher and the reduction of power consumption results in more carbon reduction in the usage stage and also causes a much lower battery capacity requirement.

When looking at the emission reduction in different stages, there are three parts: battery manufacturing stage, usage stage, and end-of-life stage. For PHEVs and EREVs, the carbon reduction in the usage stage accounts for the vast majority, followed by the battery manufacturing and the end-of-life stages. As for BEVs, the carbon reduction percentage in the battery manufacturing stage is the same as in the usage stage. However, the amount of reduction is not the same. The reduction is only 0.04 t (5.1%) in the battery manufacturing stage for BEV100 and 0.35 t (5.1%) in the usage stage. However, the emission reduction increases at a different rate as the driving range increases. For BEV1000, the reduction in the battery manufacturing stage of 0.52 t (5.7%) is even higher than that in the usage stage of 0.45 t (5.7%).

The representative A-class SUVs are selected and it is found that the power consumption factor influences the carbon emission of all stages except manufacturing (excluding battery). In addition, it is noticed that the carbon reduction amount and its ratio of EREVs are higher than most BEVs, except BEV1000. The main reason for this could be that the effect of power consumption on carbon emission is related to the curb weight, so the effect on EREVs is higher, for they are relatively heavier compared to other vehicles in the same class.

### 3.3. Sensitivity Analysis of Battery Power Density

Battery power density is a key factor influencing the carbon emission of all stages and is improving quickly. There are two main types of battery used in PHEVs, EREVs, and BEVs, which are lithium iron phosphate (LFP) batteries and lithium nickel cobalt manganese oxide (NCM) batteries. The power density of these batteries is not the same, but overall, an average improvement of 60% from 2020 to 2030 is predicted in the road map 2.0 [23]. The relationship between power density and the LCA-based emission is nonlinear because the power density also influences other parameters of representative models, which is shown in Equation (6), and impacts the emission indirectly. Figure 3 shows the comparative difference in carbon emission when the average power density of the battery is 10% higher or lower. The representative A-class SUVs are selected to compare the reduction effect in different stages.

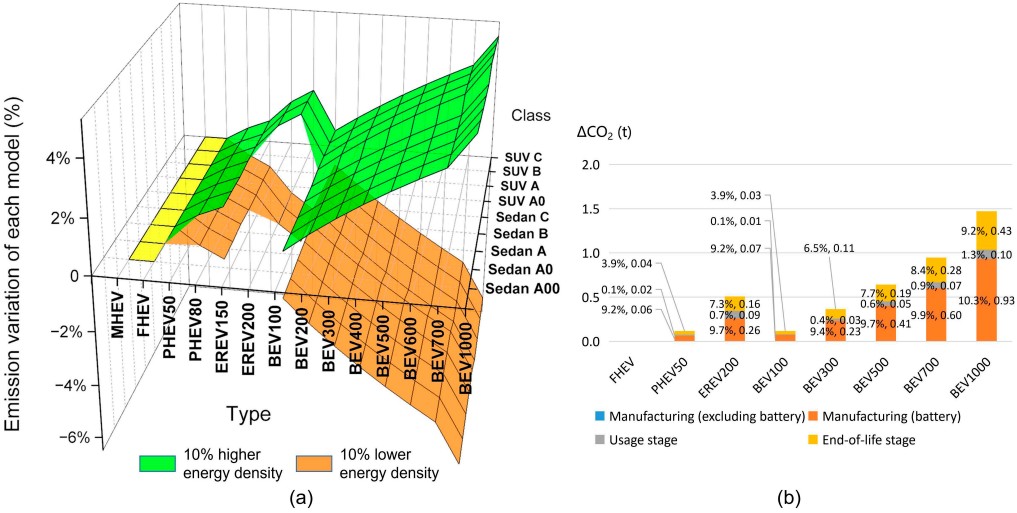

**Figure 3.** The influence of battery power density on each type of vehicle. (**a**) Sensitivity analysis of battery energy density. (**b**) The emission distribution of different sources (A-class SUV). The numbers are emission reduction amount according to $\Delta CO_2$ (t) and the percentage is the rate of emission in the corresponding stage (reduced carbon emissions/original carbon emissions).

The results show that the results of a 10% change in carbon emission are not symmetrical. If the power density fails to reach the goal in the road map, the emission of vehicles would be much lower compared to the scenario of 10% higher, which also shows that the means of reducing carbon emissions through energy density will become increasingly difficult as it develops, because upgrading energy density is becoming more and more difficult, and its impact on vehicle carbon emissions is becoming less and less.

When comparing the effect in different stages, it can be seen that the power density widely affects the emission of NEVs in all stages. The main stage for emission reduction is battery manufacturing, followed by the end-of-life stage and usage stage. The effect in the usage stage is the lowest because it is only caused by the indirect influence that higher density results in fewer batteries, resulting in lower curb weight and lower power consumption. For BEVs, the carbon reduction rises dramatically with increasing driving range. The reduction is only 0.07 t (9.2%) in the battery manufacturing stage, 0.01 t (0.1%) in the usage stage, and 0.03 t (3.9%) in the end-of-life stage for BEV100. However, for BEV1000, it is 0.93 t (10.3%) in the battery manufacturing stage, 0.10 t (1.3%) in the usage stage, and 0.43 t (9.2%) in the end-of-life stage, more than 10 times the amount. Power consumption does not significantly affect the emission of PHEVs, but for EREVs, there is a reduction of 0.26 t (9.7%) in the battery manufacturing stage, 0.09 t (0.7%) in the usage stage, and 0.16 t (7.3%) in the end-of-life stage.

When comparing the effect in different vehicle types, the effect of battery energy density is strongly related to the battery capacity. All NEVs benefit from this factor a

lot but not FHEVs or PHEV50. The carbon emission reduction as well as the reduction ratio increase with driving range. The importance of energy density is significant for long-driving-range BEVs. Additionally, the ratio of carbon reduction is similar between PHEVs, EREVs, and BEVs with the same driving range.

### 3.4. The Comparative Sensitivity Analysis

The main purpose of carrying out a comparative sensitivity analysis is to compare the actual effect of different factors on LCA-based emission, considering the development of these factors and the influence of these factors. The first step is to unify these influencing factors under the same scale according to their development speed. To unify all factors, the range of each factor's changes from 2020 to 2030 is regarded as 100%, then the influence on all representatives of each factor alone is calculated, so that the results can be comparable. To better compare the influence between different factors and different representative models, the total amount of carbon reduction is calculated, as well as the contribution rate of the carbon reduction. The results are shown in Figure 4.

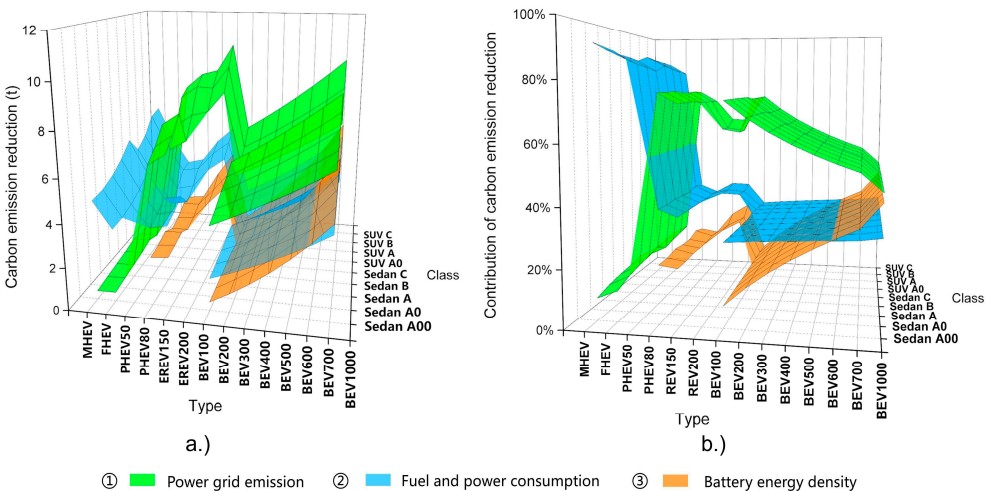

**Figure 4.** The carbon reduction caused by different factors: (**a**) the amount of the average carbon reduction for each vehicle; (**b**) the contribution rate of different factors.

Every data point in Figure 4a stands for the carbon emission caused by a single factor for a representative model. Data points in Figure 4b stand for the ratio between the carbon reduction caused by a single influencing factor and the total amount of the reduction between 2020 and 2030 caused by all factors combined, for a representative model. Since the impact of each factor on carbon emissions is complex, the sum of the contribution rates caused by each factor does not equal the simultaneous effect caused by all three factors. It can be seen that there is a significant difference in carbon reduction between different power types and different influencing factors. It can be inferred that:

Power grid emission: The most important contributor to the carbon emission reduction of all vehicles except HEVs. The contribution rate to the emission reduction effect of BEVs decreases with the increase in driving mileage, from 74% to about 46%. Meanwhile, the power grid emission has less impact on HEVs.

Battery energy density: The emission reduction effect increases rapidly with the battery capacity, from 7.4% to 42.9%, but is always lower than the power grid emission factor.

Power consumption and fuel consumption: This is an important factor that influences most vehicles. The reduction contribution of most vehicles is higher than battery energy density, except for BEVs with electric driving ranges of over 600 km.

In general, the influence of power grid emissions has the largest impact on all vehicle types except HEVs. The influence of fuel and power consumption is the second most important factor for most of the NEVs except vehicles with driving ranges over 600 km. Although the energy density of batteries has improved quickly, the actual effect is not as

great as that of power consumption. The ratio of the influence caused by different factors can be concluded as follows (① is power grid emission, ② is power consumption, ③ is battery energy density):

For HEV models: ② > ① > ③;

For PHEV and EREV models: ① > ② > ③;

For BEV models with a driving range of less than 600 km: ① > ② > ③;

For BEV models with a driving range of less than 600 km: ① > ③ > ②.

## 4. Conclusions

Many factors influence the carbon emission of vehicles, some are parameters of vehicles, and other influencing factors are not vehicle performance parameters. This research studies three of these factors that have changed rapidly and also have a big impact on life cycle carbon emissions, which are power grid emissions, battery energy density, and power consumption. To better compare the influence, both separate and comparable sensitivity analyses are conducted.

The results show that the power grid emission universally affects the carbon emission of all models in all stages, but the effects are different, mainly the emission caused by power consumption and carbon emission in the manufacturing stage. All NEVs benefit from power grid emissions a lot, but not ICEVs or HEVs. In general, the power grid emission has a higher effect on NEVs because they benefit in both manufacturing and usage stages. It can also be seen that the carbon emission reduction ratio of all representative models is similar because of the same reason. The relationship between power grid emissions and the carbon emission of vehicles is linear.

The battery energy density mainly influences the carbon emission caused by power consumption, and the reduction rate is strongly related to the battery capacity. The ratio of carbon reduction is similar between PHEVs, EREVs, and BEVs with the same driving range. When analyzing the source of the carbon reduction, it is clear that most of the reduction occurs in the manufacturing stage and end-of-life stage or, more precisely, in the battery manufacturing and recycling stages. The relationship between battery energy density and carbon emission of vehicles is nonlinear, and the means of reducing carbon emissions through energy density will become increasingly difficult as it develops.

Power consumption influences the carbon emission of all NEVs. This factor mainly affects the emission in the usage stage and the ratio of carbon reduction is mainly related to the power consumption of the vehicle itself. Due to this, the carbon reduction ratio of PHEVs and EREVs, which have higher curb weight and higher power consumption, is higher than that of BEVs with the same or even higher driving range. The biggest contribution of carbon reduction is in the usage stage, followed by battery manufacturing and the end-of-life stages. The manufacturing (excluding battery) stage is not influenced by the power consumption factor.

When comparing these influencing factors, it can be seen that the effect of different factors on different representative models is different and complicated, but in general, the power grid emission is the most influential factor for all vehicles excluding HEVs. Fuel and power consumption are considered together when conducting the comparative sensitivity analysis and they are the second most influential factor for most NEVs, except for BEVs with driving ranges over 600 km. The factor of battery energy density is the least influential factor for most of the vehicles, and the significance of this factor is higher than that of power consumption only if the vehicle is a BEV with a driving range higher than 600 km.

The results of sensitivity analysis not only show the effect on different vehicles but also the effect on different stages. It can be seen that the power grid emission mainly influences the carbon emission of the manufacturing stage, including manufacturing of the battery and the rest of the vehicle. Power consumption mainly affects the emissions of the usage stage. The battery energy density mainly affects the battery manufacturing and usage stages.

It can be concluded from the results of sensitivity analysis that, in general, PHEVs and EREVs benefit the most when the three factors are improved in terms of carbon reduction. Although carbon emissions of BEVs rise with driving range, the effect of carbon reduction of BEVs also rises with driving range. So, long-driving-range BEVs will become more and more environmentally friendly and their carbon emissions will become closer to those of BEVs with low driving range. The carbon emission of EREVs is falling the fastest among all vehicles and they will become a good solution when considering both LCA-based carbon emissions and driving range anxiety because their carbon emission in all stages responds to all factors well.

The influence of each influencing factor on the emission reduction effect of different models is different. The main reason for the low carbon emission reduction potential of ICEVs and HEVs is that the emission reduction effect of the three influencing factors cannot be reflected in these power systems.

For different power systems, the level of sensitivity can also determine the priority of technology research and development, and the influencing factors with high sensitivity should also become a bigger concern. For example, although vehicle manufacturers are not able to change the average power grid emission, still, the best means of reducing carbon is promoting the usage ratio of clean electricity, especially in factories. Both battery energy density and power consumption technology should be improved when seeking lower LCA-based carbon emissions. Except for BEVs with a driving range over 600 km, the power consumption should be given more attention.

## 5. Policy Suggestions and Limitations

### 5.1. Policy Suggestions

The automobile industry has a long industrial chain, and its life cycle carbon emissions are closely related to many links in the supply chain. In recent years, China has attached great importance to the control of carbon emissions, and many of its measures have been related to the automobile industry chain.

In terms of power grid policies, on the one hand, China has collected a renewable energy price surcharge from the electricity sales price as a renewable energy development fund [47] since 2006, and set subsidies for solar and wind power generation until 2020. On the other hand, all provinces have formulated renewable energy consumption indicators [48]. By 2030, renewable energy in each province should reach 40% of the total electricity consumption, of which nonhydropower should reach 25.9%.

As the power grid is the most effective emission reduction factor, it is the top priority to speed up development progress as much as possible. The cost of photovoltaic power generation is lower than that of coal–thermal power generation. Therefore, it is more important to build various new facilities in the power grid system to ensure that the power system can still operate stably when the proportion of these kinds of unstable renewable energy is relatively high, rather than simply tightening the management policy.

In terms of carbon emission management policies of relevant enterprises in the supply chain, China has now established a carbon emission market to downgrade the carbon emissions of enterprises. At present, industries directly related to passenger vehicles include the power industry, oil industry, and metal smelting industry. There are also individual carbon trading market pilot areas that include automobile manufacturers. These carbon markets can be seen as both punishments for high-emission technologies and financial support for low-emission technologies.

However, as far as the current situation is concerned, even without technological improvement in the whole supply chain, the carbon emissions of vehicles will still reduce quickly. In addition, many Chinese automobile manufacturers as well as foreign manufacturers are promoting the clean use of electricity in the production process. By building renewable energy power stations or purchasing renewable energy power for the production process, they can achieve zero emissions in manufacturing factories. However, this action was not required by the policy. In the short term, requiring automobile manufacturers to

fully convert to renewable energy power through the carbon market is a measure that can be quickly realized and has a significant effect, especially for enterprises producing traditional diesel vehicles. However, if we want to comprehensively reduce carbon emissions in the entire industrial chain, we need to include all industries in the entire industrial chain in the scope of carbon market policy management, which is very difficult and only leads to a small reduction of carbon emissions, and therefore is not a very urgent demand.

At this time, regarding the double-credit policy, the standard of power consumption and fuel consumption can be raised to require enterprises to apply advanced technology. Referring to the logic of the current double-point regulation, a manufacturer is taken as the object of assessment, the maximum limit value is set for all models, and the average target value is set for assessment. The carbon emissions related to the automobile industry have also dropped to a lower level, and the scope for further decline is limited. Fuel vehicles and new energy vehicles can have a relatively uniform standard of difficulty and there is no longer a bias against new energy vehicles. At present, the relevant government departments and research institutions of China's automobile industry are studying the method of combining the dual-point regulation with the carbon trading market, the purpose of which is to convert fuel consumption and electricity consumption into carbon emission indicators for the assessment of enterprises. However, according to the current information, the carbon emissions in the use phase of the past decade will not be included in the assessment scope.

To sum up, the route of reducing carbon emissions in the automobile industry should be divided into two stages. In the short term, the electrification of passenger vehicles should be promoted, which benefits from the emissions reduction of the power grid and, in this stage, the green transition of power generation is fast and spontaneous in terms of cost. The technology progress is fast during this stage, since new energy vehicles are newly implemented in China. In the long term, the emissions of the power grid will be constantly reduced till achieving carbon neutral. With more industries in the industrial chain included in the carbon market in China, the passenger vehicle industry will enter the final stage of deep decarbonization.

### 5.2. Limitations

This research mainly studies the LCA-based carbon emission of all types of vehicles. The main purpose of this research is to compare the effects of three main influencing factors on all stages of the life cycle. Both the sensitivity of each factor for each vehicle and the change in emissions under the influence of various factors' progress from 2020–2030 are analyzed. However, the factors considered in this research are limited and the variation is not studied detail. To give more accurate advice to companies in the supply chain, the sensitivity analysis can be conducted with LCA-based emissions divided into more subdivided stages so that the suggestion of technology research and development will be more specific for all suppliers and manufacturers. Additionally, it is found that the relationship between power consumption, battery energy density, and carbon reduction is not linear, and the effect of their influence deserves further in-depth analysis. A prediction of the emissions in the mid-to-long term is urgently needed, especially after China proposed to achieve carbon neutrality in 2060. More research needs to be carried out in future.

**Author Contributions:** Conceptualization, F.Z. and Z.L.; methodology, H.Z.; validation, F.Z. and Z.L.; formal analysis, Z.L.; investigation, H.Z.; resources, F.Z. and H.H.; data curation, H.H.; writing—original draft preparation, H.Z.; writing—review and editing, H.Z.; visualization, H.Z.; supervision, F.Z.; project administration, Z.L.; funding acquisition, Z.L. All authors have read and agreed to the published version of the manuscript.

**Funding:** This research was funded by the National Natural Science Foundation of China, grant number 52272371.

**Institutional Review Board Statement:** Not applicable.

**Informed Consent Statement:** Not applicable.

**Data Availability Statement:** Data is contained within the article.

**Conflicts of Interest:** The authors declare no conflict of interest. The funders had no role in the design of the study; in the collection, analyses, or interpretation of data; in the writing of the manuscript; or in the decision to publish the results.

## Nomenclature

| Abbreviations | Definitions |
| --- | --- |
| GHG | Greenhouse gas |
| NEV | New energy vehicle |
| LCA | Life cycle assessment |
| OEM | Original entrusted manufacture |
| ICEV | Internal combustion engine vehicle |
| HEV | Hybrid electric vehicle |
| MHEV | Mild hybrid electric vehicle |
| FHEV | Full hybrid electric vehicle |
| PHEV | Plug-in hybrid electrical vehicle |
| EREV | Extend range electric vehicle |
| BEV | Battery electrical vehicle |
| CATC | China Automotive Test Cycle |
| WLTC | Worldwide Harmonized Light Vehicles Test Cycle |
| CD | Charge depleting |
| CS | Charge sustaining |
| NHTSA | National Highway Traffic Safety Administration |
| SUV | Sports utility vehicle |
| LFP | Lithium iron phosphate |
| NCM | Lithium nickel cobalt manganese oxide battery |
| $C_i$ | The battery capacity of model $i$ (kWh) |
| $E_{LCi}$ | Life cycle emissions of vehicle model $i$ (kgCO$_{2e}$) |
| $E_{VMi}$ | Emissions of vehicle manufacturing of vehicle model $i$ (kgCO$_{2e}$) |
| $E_{BM}$ | Emissions during battery manufacturing (kgCO$_{2e}$) |
| $E_{BE}$ | Emissions caused by the battery installed in vehicle model $i$ during the end-of-life stage (gCO$_{2e}$/kg) |
| $E_{VEi}$ | Emission in the end-of-life stage of vehicle model $i$ (kgCO$_{2e}$) |
| $EC_i$ | The electric consumption of model $i$ (kWh/100 km) |
| $FC_i$ | The fuel consumption of model $i$ (L/100 km) |
| $FE$ | The life cycle emission of fuel consumption (CO$_{2e}$/L) |
| $GE$ | Power grid carbon emission (kg/kWh) |
| $M_i$ | Curb weight of vehicle $i$ (kg) |
| $M_{Bi}$ | The mass of the power battery of model $i$ (kg) |
| $M_{Ri}$ | The mass of model $i$ excluding batteries (kg) |
| $R_u$ | Driving mileage of vehicle during usage stage (100 km) |
| $R_i$ | The electric driving range of model $i$ (100 km) |
| UF | Utilization factor (%) |
| $\omega$ | The predicted average battery density (Wh/kg) |

CO$_{2e}$ stands for carbon dioxide equivalent.

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
