# Peer review of "Life Cycle Emissions of Passenger Vehicles in China: A Sensitivity Analysis of Multiple Influencing Factors"

_sustainability, doi:10.3390/su15064854_

Round 1

Reviewer 1 Report

Congratulate the authors for the work done, it is a current issue and a great contribution to research.

The analysis of the works or investigations should be improved with respect to the previous and current theoretical and empirical background on the subject.

References to similar work relevant to the research should be included.

Author Response

Comment 1: The analysis of the works or investigations should be improved with respect to the previous and current theoretical and empirical background on the subject. References to similar work relevant to the research should be included.

Response: Thanks for your patient reading and your comments. It is noticed that my introduction of this research field is not complete. The research is conducted with my new developed calculation model. The main difference between this research and others is that instead of calculating and comparing the emissions base on single vehicle type, various models with significant differences are compared to get more comprehensive and precise conclusions. Such information should be describe clearer indeed. Content in the Introduction chapter is expanded and references are added,marked in red.

Reviewer 2 Report

The paper presents studies on the life cycle carbon emissions of passenger vehicles in China. Authors developed calculation model that helps to calculate and compare the effect of carbon emission reduction on vehicles. Sensitivity analysis was also performed. 

The paper is well structured and makes an important contribution to understanding the impact of CO2 emissions from civil vehicles on the atmosphere over its lifetime.

The following faults were perceived in the work:
1. Incorrect numbering of figures. Figure 1 does not appear anywhere; the authors should check whether they forgot to insert it.
2. the Table Abbreviation (p. 5) should be moved to the beginning of the paper and named as List of symbols, and completed with the other designations appearing in the paper that were not in the "Table. Abbreviation".
3 The quality of the drawings should be improved. In their current form, they are hardly legible.
4) Please check and correct the contents of the Funding and Data Availability Statement on page 18.

Author Response

Comment 1: Incorrect numbering of figures. Figure 1 does not appear anywhere; the authors should check whether they forgot to insert it.

Response: Thanks for your patient reading and your comments. These problems are noticed and corrected. Figure 1 in the original manuscript is deleted and other figures are rearranged.

Comment 2: The Table Abbreviation (p. 5) should be moved to the beginning of the paper and named as List of symbols, and completed with the other designations appearing in the paper that were not in the "Table. Abbreviation".

Response: Thanks for your comments. A nomenclature table is added at the beginning, all abbreviation and variables in this research are listed in this table.

Comment 3: The quality of the drawings should be improved. In their current form, they are hardly legible.

Response: Thanks for your comments. The diagrams in the original manuscript are clear, but the clarity decreases after being inserted into the manuscript. To fix the problem, all figures in this paper are redrawn and rearranged in the manuscript, to make sure they are still legible after zooming in. Full size pictures are also added to the supplementary materials.

Comment 4: Please check and correct the contents of the Funding and Data Availability Statement on page 18.

Response: Thanks for your comments. There are some error in these two parts indeed, corresponding revise has been made.

Reviewer 3 Report

In the reviewer’s opinion, the paper could have been more interesting and better organised. In general, the overall contribution remains scientifically poor and technically questionable. In more detail, the paper’s title is quite long, whilst its Abstract should have avoided the use of acronyms, which reduce its readability. The keyword list needs to be improved. Section 1 cites some references, but it does not provide a sufficiently exhaustive overview and critical discussion of the state of the art of the related literature. As further remark, the use of itemised list reduces the readability of Section 1. Section 2 should have addressed more details regarding the considered models and tools; in particular, it does not consider the robustness and reliability issues, due for example to uncertainty and disturbance effects, as well as the model-reality mismatch. This point is fundamental when the reliability and robustness features of the proposed solutions have to be verified and validated with respect to real engineering and safety critical systems. Therefore, the effectiveness of the methodology proposed in Section 2 remains unclear and questionable. The authors should have helped the reader to understand the novelty issues of the developed scheme. Due to these flaws, the results considered in Section 3 do not help the reader to understand the effectiveness and the efficacy of the proposed solutions. The authors reported many pictures and tables. However, more effective metrics and performance indices should be exploited to assess the advantages of the developed techniques. Finally, Sections 4 and 5 are very long. Section 4 should have ended the manuscript. They do not suggest effective open problems and future issues that could require further investigations. Section 6 should have been included in Section 5. On the other hand, the use of acronyms should have been avoided also here, as it should remain a stand-alone part of the manuscript.

Author Response

Comment 1: The paper’s title is quite long, whilst its Abstract should have avoided the use of acronyms, which reduce its readability. The keyword list needs to be improved.

Response: Thanks for your patient reading and your comments The reason why the title is longer now is that we want to explain the difference between this study and other life cycle carbon emission studies, but it can be more simplified. Now the title has been modified to simplify it a lot. Modifications are made in the abstract part for better readability, and all acronyms are removed. Keywords are also been revised, marked in red.

Comment 2: Section 1 cites some references, but it does not provide a sufficiently exhaustive overview and critical discussion of the state of the art of the related literature. As further remark, the use of itemised list reduces the readability of Section 1.

Response: Thanks for your comments. The research is conducted with a own built calculation model (LCARM). The main difference between this research and others is that instead of calculating and comparing the emissions base on single vehicle type, various models with significant differences of vehicle class, power system, and electric driving range are compared to get more comprehensive and precise conclusions. Such information should be describe clearer indeed. Content in the Introduction part is expanded, a literature review is added, and corresponding references are added, marked in red.

Comment 3: Section 2 should have addressed more details regarding the considered models and tools; in particular, it does not consider the robustness and reliability issues, due for example to uncertainty and disturbance effects, as well as the model-reality mismatch. This point is fundamental when the reliability and robustness features of the proposed solutions have to be verified and validated with respect to real engineering and safety critical systems. Therefore, the effectiveness of the methodology proposed in Section 2 remains unclear and questionable. The authors should have helped the reader to understand the novelty issues of the developed scheme. Due to these flaws, the results considered in Section 3 do not help the reader to understand the effectiveness and the efficacy of the proposed solutions. The authors reported many pictures and tables. However, more effective metrics and performance indices should be exploited to assess the advantages of the developed techniques.

Response: Thanks for your comments. Indeed, a newly established model should be demonstrated to discuss the model-reality match and robustness. This study is based on the model in the previous published research. More details of the model have been proposed in the previous paper. In this study, the establishment of the model is not the main part of the elaboration. I only briefly introduce the calculation methods, necessary assumptions and partial input data. The purpose of this study is to discuss the three parameters that have a great impact on the carbon emissions of vehicle life cycle and the average level changes rapidly. The purpose of this sensitivity analysis is, on the one hand, to intuitively express the effect of various influencing factors, and on the other hand, to calculate the error when the three influencing factors fluctuate higher or lower than the expected value in 2030. These possible errors have already been reflected in the "results" chapter.

Based on your suggestion, I found that there are other parameters that may affect the calculation results of carbon emissions. These parameters neither have a certain development trend with the progress of technology, nor widely distributed among different models in the market. However, once its average value changes, the result will still be different from the real situation. I found that the two parameters of driving range and vehicle curb weight in the use stage may have a great impact on the carbon emissions of vehicle models. The corresponding analysis was added to the chapter "results" and marked in red, but because it is not the main research goal, it has not been studied in detail and is not shown in chart or figures.

Comment 4: The authors reported many pictures and tables. However, more effective metrics and performance indices should be exploited to assess the advantages of the developed techniques.

Response: Thanks for your comments. All the figures have been redrawn to make them more legible. Some explanations are added to the figure to show the meaning of coordinates and surfaces. However, the main purpose of this research is to illustrate compare the carbon emission reductions of all types of vehicles on the same figure. Therefore, although the surface graph is somewhat difficult to understand, it is still the most intuitive representation at present. Further instructions are added to the article and marked in red.

Comment 5: Sections 4 and 5 are very long. Section 4 should have ended the manuscript. They do not suggest effective open problems and future issues that could require further investigations. Section 6 should have been included in Section 5. On the other hand, the use of acronyms should have been avoided also here, as it should remain a stand-alone part of the manuscript.

Response: Thanks for your comments. This issue is not noticed when drafting, policy recommendations are an important part of this study. I hope to find out the key factors that have a greater impact on the carbon emissions of various models based on the comparison results of the influencing factors. Obviously, the policies that can improve this factor need more attention from the government. The policy recommendations really are what I need to mention, please understand me. But the length of chapters 4, 5 and 6 is too long indeed, a substantial reduction can be done. According to your suggestions, I have combined chapters 5 and 6 and deleted half of the contents. At present, the fourth chapter is mainly about sensitivity analysis and comparison results. The fifth chapter briefly puts forward policy recommendations. These revision are not marked red, because the revision is of this part is mainly deletion, and nearly all the contents have been modified. If you still think it is necessary to further simplify the content, I think more deletions are possible.

Round 2

Reviewer 3 Report

The paper may be accepted in current form.